# The Impacts of Fiscal Subsidies on the Carbon Emissions of Mining Enterprises: Evidence from China

**DOI:** 10.3390/ijerph192316256

**Published:** 2022-12-05

**Authors:** Wei Dai, Xuefang Zhang, Chaohui Xu

**Affiliations:** 1School of Economics and Management, Hubei Polytechnic University, Huangshi 435003, China; 2School of Economics and Management, Hubei University of Science and Technology, Xianning 437100, China

**Keywords:** fiscal subsidies, mining enterprises, carbon emission, green-technology innovation

## Abstract

Lowering carbon output and reducing emissions have been worldwide concerns as global warming and environmental pollution worsen. Governments play a key role in driving corporate action related to carbon and emission reduction. In this paper, mining companies listed in China’s A-share market were taken as samples to analyze the impacts of fiscal subsidies on the carbon emissions of mining enterprises through empirical tests. These findings demonstrated that fiscal subsidies could substantially lower corporate carbon emissions by incentivizing and enhancing their green-technology innovation. Financing constraints provided no prominent mediator effects between fiscal subsidies and carbon emissions, and these subsidies failed to considerably relieve their financing constraints in order to restrain carbon emissions. These results indicate that government policies on fiscal subsidies could represent significant guidance for corporate low-carbon and environmental-protection efforts, thereby providing empirical evidence for governmental environmental-protection policies.

## 1. Introduction

As a result of surging carbon emissions, global warming has become a vital global environmental issue. With rapid development in China’s economy, CO_2_ emissions have increased dramatically. China’s mining industry is a fundamental part of the national economy and a key source of energy, industrial raw materials, and agricultural production means. It also plays an important role in securing state resources, promoting national economic growth, and facilitating regional economic construction. However, this continuous industry development has been accompanied by soaring energy consumption and CO_2_ emissions. This indicates that focusing on reduction of CO_2_ emissions in China’s mining industry is of great importance for achievement of overall emission-reduction targets. Based on this background, this paper focuses on the impacts of fiscal subsidies on the CO_2_ emissions faced by China’s mining industry.

Previous research and practical experience have both demonstrated that fiscal policies play a vital role in addressing climate change and promoting low-carbon economic development, meaning these policies are indispensable in carbon-emission control. The studies of López et al., Martínez-Zarzoso, and Ang [1,2,3] found that fiscal expenditures have not only a direct impact on carbon emissions but also an indirect impact on urbanization, industrial structure, and foreign trade. An inverted U-shaped relation (ascending first, then descending) exists between fiscal expenditures and carbon emissions. Adhikari and Agrawal [4] showed that, in fiscal expenditure structure, increasing fiscal expenditures for science and technology helps to lower carbon-emission intensity. Shan et al. [5] found that increasing fiscal expenditures for environmental protection effectively promoted carbon-emission reduction. In different economic development stages, there is a U-shaped relationship between win–win ability and the economic development stage [6]. Dogan and Seker [7] suggested that increasing public expenditures for noneconomic, nonproductive purposes helps to lower regional carbon emissions. These researchers analyzed the impacts of diverse fiscal expenditures on carbon emissions. 

Some researchers have analyzed carbon-emission reduction effects of green fiscal policies. Lin and Zhu [8] found that construction of all-around demonstration cities in energy-conservation and emission-reduction policies lowered the carbon emission levels of cities remarkably well. Other researchers have tested the impact of three environmental fiscal policies—environmental expenditures, environmental revenues, and environmental subsidies—on the carbon emissions of East, Central, and West China from empirical perspectives, finding the that impacts of these expenditures on those areas’ carbon emissions were insignificant. Niu et al. [9] found that revenues improved the carbon emissions in East China, verifying the so-called Green Paradox; these subsidies have restricted the carbon emissions in these areas. In addition, Li et al. [10] researched the impact of government subsidies on low-carbon supply chains. Ma et al. [11] studied the impacts of government interventions on carbon emissions. Ma et al. [12] analyzed the impacts of government subsidies on lowering carbon consumption. Elhedhli and Merrick [13] investigated inhibition of green supply-chain network designs on carbon emissions. Dietz et al. [14] explored the governing effects of household actions on carbon emissions. Cavallaro et al. [15] assessed inhibition of carbon emissions via road-pricing plans. Wang and Feng [16] demonstrated that the driving factors in the mining industry’s various subsectors are quite different when it came to carbon emissions. They also [17] found that at the global level, both nurturing pressure and unemployment had negative effects on CO_2_ emissions.

In conclusion, though some research has focused on the impacts of different fiscal policies and measures on carbon emissions, few researchers have analyzed the impact of fiscal subsidies on carbon emissions. Currently, the Chinese government attaches great importance to low-carbon economic growth and has highlighted the roles of fiscal policies and instruments in that growth. The study in this paper sought to determine whether fiscal subsidies—one of several important fiscal policies and measures—have been effective in controlling carbon emissions, and to elucidate the inherent mechanism.

The rest of this article is organized as follows: Section 2 is a literature review and hypothesis proposal; Section 3 is methodology and data; Section 4 follows with empirical results; Section 5 provides discussion and policy implication; and finally, Section 6 summarizes this full text and puts forward future prospects.

## 2. Literature Review and Hypothesis Proposed

Fiscal subsidies inhibit the carbon emissions of mining enterprises through relief of their financing constraints. Theoretically, they can relieve financing constraints through two paths: direct and indirect. Regarding the direct path, Howell [18] showed that governments could provide enterprises with fiscal subsidies, directly supplementing corporate financing gaps. These subsidies are undoubtedly necessary for enterprises that face serious financing constraints. Regarding the indirect path, fiscal subsidies can help enterprises obtain more funds and lower their financing costs, indirectly relieving their financing constraints. These subsidies provide certification effects, meaning the industry or services will be recognized by governments if fiscal subsidies are obtained [19]. It is a positive signal, showing that these enterprises have promising prospects and helping them attract funds from potential social investors or lenders to relieve their difficulties and high expenses [20,21]. Some researchers have provided empirical support for this, demonstrating that fiscal subsidies had significant signaling effects for banks and social investors via aid of subsidized enterprises in acquiring bank loans and funds raised from capital markets [22]. As invisible guarantees from governments, fiscal subsidies lower the banking risk ratings of these enterprises, reducing their debt financing costs [23]. Colombo et al. [24] found that these subsidies substantially lower investment–cash flow sensitivity of small enterprises; they studied the effects of fiscal subsidies on 288 Italian high-tech enterprises and implied that those subsidies effectively relieved the enterprises’ financing constraints. Relevant research indicated that financing constraints could also inhibit carbon-emission reduction. Managers’ initiatives in energy conservation and emission reduction can be enhanced—or enterprises motivated to focus on green-technology innovation and environmental-protection investment—if their financial constraints are relieved. This, correspondingly, could enhance carbon-emission reduction.

Fiscal subsidies inhibit carbon emissions by incentivizing corporate green-technology innovation. As a key measure to regulate markets, fiscal subsidies profoundly influence research and development behaviors of enterprises. Most research has shown that these subsidies facilitate corporate investments in research and development through resource attributes and signal transmission. Regarding resource attributes, subsidies can promote investment in research and development through enterprises within strategic emerging industries. On the other hand, these subsidies can function as a signal transmission to help acquire resources and support from stakeholders for direct innovation outputs. However, more researchers found that fiscal subsidies were able to increase the proportions of enterprises engaged in research and development activities based on their empirical analyses. A few suggested that the impact of fiscal subsidies on corporate investments in research and development was nonlinear, and that if the subsidies exceeded critical values, they could even demonstrate crowding-out effects [25,26]. Mining enterprises mainly utilize three methods—curtailing production, purchasing energy-saving equipment, and conducting green-technology innovation—to reduce carbon emissions. Curtailing production is analogous to giving up eating for fear of choking, and updating equipment can be extremely costly. Therefore, green-technology innovation is a crucial measure to reduce carbon emissions, and improving green-technology innovation ability is a central driving force for achieving carbon-emission reduction goals. Relevant research has demonstrated that green-technology innovation can lower corporate carbon emission levels [27] and effectively promote carbon-emission responsibilities. Additionally, independent technological innovation can have a remarkable effect on carbon-emission reduction [28]. Of course, apparent regional differences can be found; independent research and development has impacted carbon-emission reduction in East and West China, but the impact in Central and Northeast China has been insignificant. Other research has implied that the effects of technological innovation on carbon emissions are nonlinear and that the relationships between factors are influenced by income levels. In high-income areas, technological innovation produced remarkable carbon-emission reduction effects, while the effects in low-income areas were insignificant.

A mining enterprise possesses unique features that differentiate it from other enterprises. Production from mining enterprises strongly relies on natural resources and could pollute or damage natural environmental elements, such as air, water, and soil. In fact, while the mining industry has a long history in China, its extensive development over the long term has resulted in a series of social problems, including severe pollution (waste gas, water, and residuals) and inadequacy of reserve resources. These social problems, caused by extensive development, accelerated the pace of China entering an era with high risks of environmental pollution. Incidents of environmental pollution, ecological damage, and pollution range extension induced by mining in specific areas have attracted significant attention from the public, putting pressure on mining enterprises to find solutions to these environmental problems. Therefore, it has become dramatically important to empirically study the carbon emissions of mining enterprises.

In short, green-technology innovation effectively helps enterprises reduce their carbon emissions, and, as such, is an important measure for carbon-emission reduction. Some empirical tests have verified notable carbon-emission reduction effects of fiscal subsidies in carbon-emission governance, expanding the governance mechanism for carbon emissions. Additionally, the mechanism through which fiscal subsidies impact carbon emissions has been verified with financing constraints and green-technology innovation, revealing the so-called black box between fiscal subsidies and the carbon emissions from mining enterprises. Based on the analysis above, the following hypothesis was proposed in this paper:

Fiscal subsidies obtained by mining enterprises correlate significantly with their carbon emissions: that is to say, the higher the fiscal subsidies for mining enterprises, the lower their carbon emission levels. 

## 3. Methodology and Data

### 3.1. Data Source

Chinese-listed companies have generally implemented the new accounting standard as of 1 July 2007. In this paper, listed companies in China’s mining industry from 2007 to 2020 were chosen as initial research samples. Specific sectors included non-metal mineral mining and processing, non-metal mineral manufacturing, ferrous metal mining and processing, ferrous metal smelting and rolling, auxiliary mining activities, coal mining and washing, petroleum and natural gas extraction, petroleum processing, coking, nuclear fuel processing, nonferrous metal mining and processing, and nonferrous metal smelting and rolling. The following processes were conducted for the original data: (1) Listed companies labeled ST and *ST within the sample period were removed; (2) Samples without key index values were removed; (3) Samples with an asset-to-liability ratio >1 were sampled. Finally, 3230 sample observations from 404 companies were obtained from the Institute of Public and Environmental Affairs Database; *China Energy Statistical Yearbook*; the CSMAR Database; and annual reports of listed companies, issued by SSE and SZSE. Continuous variables at the 1% quantile were Winsorized to control influence of abnormal values. 

### 3.2. Variable Selection

Explained variable. Carbon emissions (*C*) were measured in carbon dioxide (CO_2_). The natural logarithms for the provincial-level carbon emissions of corporate domiciles were taken as key explanatory variables, referring to the practices of Ji and Yang [29] as well as Mei et al. [30].Explanatory variables. Fiscal subsidies (*Sub*) were measured by the ratio of fiscal subsidies received by a mining enterprise to that enterprise’s total assets.Controlled variables. Aside from fiscal subsidies, many factors influence carbon emissions. According to the existing literature, the impacts of corporate characteristics and governance are controlled variables. Main corporate characteristics primarily include profitability (measured with return on asset; indicated as *ROA*) [31], financial leverage (measured with asset-to-liability ratio; indicated as *Lev*) [32], corporate growth (measured with corporate asset growth rate; indicated as *Growth*) [32], and corporate size (measured with the natural logarithm of corporate assets; indicated as *Size*) [33]. Corporate governance is composed of major shareholder governance (measured with the share proportion of the largest shareholder; indicated as *Ls*) [33], executive compensation (measured with the natural logarithm of executive monetary compensation; indicated as *Ggxc*) [34], independent director governance (measured with the proportion of independent directors to total directors; indicated as *Board*) [34], and two-post executives (1 when the positions of general manager and president were served by one person and 0 for other cases; indicated as *Pt*) [34].

### 3.3. Model Specification

The following linear regression model was built for this research hypothesis: (1)$$C_{i,t}=c_{0}+\alpha_{1}Sub_{i,t}+\alpha_{i}Control_{i,t}+\sum Industry+\sum Year+\varepsilon_{i,t}$$

The explained variable in Regression Model (1) is carbon emission (*C*), the core explanatory variable is fiscal subsidies (*Sub*), *Control* is an aforementioned controlled variable, and ε is a random error. When the regression coefficient $\alpha_{1}$ in Model (1) is a significant negative; it means fiscal subsidies significantly lower CO_2_ emissions.

## 4. Empirical Results and Analysis

### 4.1. Descriptive Statistics 

Table 1 shows the descriptive statistics of the main variables. The statistical data showed that the mean value and median of the fiscal subsidies were 0.0043 and 0.0017, respectively, meaning over 50% of the mining enterprises received fiscal subsidies. The mean value and median of the carbon emissions were 10.5550 and 10.6387, respectively, and the standard deviation was 0.7512, indicating that most mining enterprises had carbon emissions and the differences were small. The mean value and median of profitability were 0.0359 and 0.313, respectively, implying that profitability was weak. The mean value and median of growth were 0.1661 and 0.0811, respectively, showing that growth was low. The mean value and median of financial leverage were 0.4869 and 0.4961, respectively, demonstrating that financial leverage was generally high. The mean value and median of corporate size were 22.6787 and 22.5589, respectively, and the standard deviation was 1.5317, indicating considerable size. The mean value and median of executive compensation were 15.1386 and 15.1275, respectively, and the standard deviation was 0.8759, implying attached importance to compensation incentives for executives. The mean value of the share proportion of the largest shareholder was 0.3914, and the median was 0.3817, indicating that the “dominant share proportion of the largest shareholder” was a common situation. The mean value of the occurrence of the general manager and president positions being served by one person was 0.1737, and the median was 0, meaning the general manager and president positions were served by the same person in fewer than 50% of the enterprises.

Table 2 shows the Pearson correlation analysis. The correlation coefficient of fiscal subsidies and carbon emissions is −0.014, meaning these subsidies can lower carbon emissions to some extent. Additionally, the correlation coefficient between the variables is lower than 0.5, indicating a low possibility of collinearity in subsequent linear regression.

### 4.2. Empirical Analysis

Specific results are shown in Table 3. When controlled variables were not considered, the regression coefficient of fiscal subsidies (*Sub*) was −4.4844 and t was −1.99; when controlled variables were considered—all of them (*ROA, Growth, Lev, Size, Board, Ggxc, Ls, Pt*) notably were fixed—while sector and year fixed effects were controlled, empirical tests were conducted on the relationship between fiscal subsidies and carbon emissions of the relevant mining enterprises. The regression coefficient of fiscal subsidies (*Sub*) was −5.6254, and t was −2.45: all significant at the 5% level. This demonstrated that fiscal subsidies had significant negative correlations with carbon emissions: the higher the fiscal subsidies, the lower the carbon emission level. The hypothesis above was verified.

### 4.3. Robustness Test

#### 4.3.1. Regression Analysis

In consideration of potential endogenous problems, the fixed-effects model was utilized, with all controlled variables (*ROA, Growth, Lev, Size, Board, Ggxc, Ls, Pt*) fixed while sector and year fixed effects were controlled. Table 4 displays the regression results. When the model was tested, the regression coefficient of fiscal subsidies (*Sub*) was −5.6254, and t was −2.45, which was significant at the 5% level. Additionally, a test for robustness standard errors was performed based on the OLS method. Specific to the impact of fiscal subsidies on carbon emissions, the robustness standard error test results showed robust heteroscedasticity, and the regression coefficients of fiscal subsidies (*Sub*) were significantly negative. The regression result was consistent with the test result above. 

#### 4.3.2. First-Difference Test

First-difference testing was conducted for Model (1) to relieve the endogenous problems that resulted from reverse causality between fiscal subsidies and carbon emissions. The first-difference model is as follows: (2)$$\Delta C_{i,t}=b_{0}+\beta_{1}\Delta Sub_{i,t}+\beta_{i}Control_{i,t}+\sum Industry+\sum Year+\varepsilon_{i,t}$$

In this model, $\Delta C_{i,t}=C_{i,t}-C_{i,t-1}$ and $\Delta Sub_{i,t}$=$Sub_{i,t}-Sub_{i,t-1}$. The impact of fiscal subsidies on carbon emissions was tested (path “△*Sub*→△*C*”) according to regression in the panel data. Table 4 gives the regression results for the model above. When controlled variables were not considered, the regression coefficient of △*Sub* was −7.4671 after the first difference was conducted. When controlled variables were considered, the regression coefficient of △*Sub* was −7.8907 after the first difference was conducted for Model (1). Both regression coefficients were negative. This showed that fiscal subsidies could significantly inhibit carbon emissions. 

#### 4.3.3. Changed-Research-Period Test

Considering the impact of the subprime crisis in 2007, data from 5 years and 10 years after the crisis were utilized to test the hypothesis of this study. When samples from 2009 to 2013 were chosen, the regression coefficient was −6.5230. When samples from 2009 to 2020 were chosen, the regression coefficient was −2.6568. Both regression coefficients were negative. Fiscal subsidies and carbon emissions still had significant negative correlations even when the research period was changed. The regression results of different periods are shown in Table 5.

### 4.4. Mediator Effect Test

The hypothesis above was verified in benchmark regression. However, the inherent mechanism of action between fiscal subsidies and carbon emissions needs to be further explored. 

#### 4.4.1. Role of Green-Technology Innovation

Green-technology innovation can effectively solve high consumption and high pollution arising from in-house irrational layouts and extensive production and development. Fiscal subsidies finance green-technology innovation in mining enterprises, guiding them to this innovation and correspondingly inhibiting their carbon emissions. Therefore, the testing path is “fiscal subsidies–green technology innovation–carbon emissions of mining enterprises”. The green-technology innovation indexes were from the CNRDS Database and Wind Database, and measured via adding 1 to the number of approved green patents for inventions in the year and using the natural logarithm.

Mediator effect models were built on the basis of Model (1) as per the mediator effect test principle. The models were as follows: (3)$$Green_{i,t}=\gamma_{0}+\gamma_{1}Sub_{i,t}+\gamma_{i}Control_{i,t}+\sum Industry+\sum Year+\varepsilon_{i,t}$$
(4)$$C_{i,t}=\lambda_{0}+\lambda_{1}Sub_{i,t}+\lambda_{2}Green_{i,t}+\lambda_{i}Control_{i,t}+\sum Industry+\sum Year+\varepsilon_{i,t}$$

The results in Table 6 show that the path “*Sub*→*C*” was first tested according to Model (1). The regression coefficient of fiscal subsidies was −5.6080, which was significant at the 5% level, indicating that subsidies could significantly inhibit carbon emissions. 

The path “*Sub*→*Green*” was tested as per Model (3). The regression coefficient was −8.4342, which was significant at the 1% level and indicated that subsidies could apparently improve green-technology innovation. 

Finally, the impacts of subsidies and green-technology innovation on carbon emissions were tested according to Model (4). The regression coefficient of the subsidies was −4.4237, which was significant at the 10% level; the regression coefficient of green-technology innovation was −0.1404, which was significant at the 1% level. 

According to the mediator effect test principle, green-technology innovation had significant mediator effects between fiscal subsidies and carbon emissions, meaning the subsidies inhibited carbon emissions via improvement of green-technology innovation of mining enterprises.

#### 4.4.2. Role of Financing Constraints

Fiscal subsidies provide enterprises with cash flow for business activities, increasing their in-house resources and relieving their financing constraints [35]. This takes place through direct and indirect provision of funds, which attracts equity investments and creditors’ rights. Mining enterprises have funds for environmental protection after their financing constraints are relieved, allowing them to purchase environmentally conscious equipment, introduce environmental technology, and employ environmental talents to reduce their carbon emissions. The SA index was utilized to reflect the degree of financing constraints. In this study, the SA index was negative and the absolute value was higher, meaning the degree was quite serious. This was determined as follows:*SA* = −0.737**Size* + 0.043**Size^2^* − 0.040**Age*.

The mediator effect models were built on the basis of Model (1) as per the mediator effect test principle. The models were as follows:(5)$${\mathrm{SA}}_{i,t}=\mu_{0}+\mu_{1}Sub_{i,t}+\mu_{i}Control_{i,t}+\sum Industry+\sum Year+\varepsilon_{i,t}$$
(6)$$C_{i,t}=\omega_{0}+\omega_{1}Sub_{i,t}+\omega_{2}{\mathrm{SA}}_{i,t}+\omega_{i}Control_{i,t}+\sum Industry+\sum Year+\varepsilon_{i,t}$$

According to the estimated results of panel data in Table 7, the path “*Sub*→*C*” was first tested according to Model (1). The regression coefficient of fiscal subsidies was −5.5732, which was significant at the 5% level and indicated that subsidies could significantly inhibit the carbon emissions of mining enterprises.

The path “*Sub*→*SA*” was tested according to Model (5). The regression coefficient of fiscal subsidies was 1.1797, which was insignificant, indicating that fiscal subsidies failed to effectively relieve financing constraints.

Finally, the impacts of fiscal subsidies and green-technology innovation on carbon emissions were tested in accordance with Model (6). The regression coefficient of fiscal subsidies was −5.0961, which was significant at the 5% level; the regression coefficient of financing constraints was −0.4044, which was significant at the 1% level. 

According to the mediator effect test principle, financing constraints had insignificant mediator effects between fiscal subsidies and carbon emissions, meaning fiscal subsidies cannot inhibit carbon emissions by relieving the financing constraints of mining enterprises.

## 5. Discussion and Policy Implication

Fiscal subsidies play a significant role in improving the carbon-emissions of mining enterprises. Designing a fiscal subsidy policy to control carbon emissions is an effective way to improve environmental efficiency of these enterprises. According to the results of this study, green-technology innovation could lower mining-enterprise carbon emission levels and effectively promote carbon-emission responsibilities. Additionally, independent technological innovation could provide remarkable carbon-emission reduction effects, which could greatly reduce pollution and damage to natural environmental elements, such as the atmosphere, water, and soil. In fact, relevant research has demonstrated that financing constraints have inhibiting effects on carbon-emission reduction. When mining enterprises’ financing constraints are relieved, managers’ initiatives in energy conservation and emission reduction can be enhanced, or mining enterprises can be motivated toward green-technology innovation and investment in environmental protection. Correspondingly, this can improve carbon emission, and incidence of environmental pollution, ecological damage, and pollution-range extension induced by mining enterprises in specific areas can be controlled. After mining enterprises receive fiscal subsidies, they are required to complete assessment targets that meet the performance expectations of local governments. At present, the number of papers and patents is still the main index in formulation of the objectives and implementation rules for the current policies. Therefore, playing the guiding role of fiscal subsidies and driving social capital into carbon-emission control of mining enterprises in order to improve environmental efficiency of the mining industry will be a major innovative measure.

To achieve carbon-emission reduction targets and promote sustainable development in China’s mining industry, three policy recommendations are proposed based on the results above. First, economic measures that are consistent with corporate incentives, such as fiscal subsidies, should be constantly utilized to intervene in green production by and operation of mining enterprises. These subsidies could be provided to facilitate green-technology innovation. Some enterprises could be encouraged to help others with joint research and development of green technology—which will present high difficulty, substantial investment, and long ROI periods—in order to completely fulfill the spillover effects of green technology. 

Second, financing constraints should be further relieved to help mining enterprises invest in research and development of environmental equipment and green technology. Though difficulty of this action and high financing costs have been worldwide problems, multiple capital markets in China have been exploring this route. Bank loans are the primary financing channel. Therefore, governments should play a guiding role and aim to provide mining enterprises with discounts for low-carbon investments, relieving those barriers.

Thus, mining enterprises must properly utilize fiscal subsidies to reduce their green development costs. Because these costs are considerable and because of short-term goals and pursuits, managers and shareholders have proven reluctant to engage in low-carbon development. Fiscal subsidies bring direct cash flow, permitting mining enterprises to specifically and efficiently utilize those subsidies to relieve their financing constraints and increase research and development input for green technology. In this way, subsidies aid mining enterprises in their efforts to achieve low-carbon targets.

## 6. Conclusions

In this paper, the impact of fiscal subsidies on the carbon emissions of mining enterprises and their inherent mechanism of action was analyzed from the perspective of green development. These findings indicated that fiscal subsidies had significant negative correlation with carbon emissions and that subsidies could considerably inhibit emissions. In addition, this study revealed the mechanism behind this impact: governments granting fiscal subsidies to urge mining enterprises to further inhibit carbon emissions through enhancement of green-technology innovation. Finally, financing constraints had insignificant mediator effects between fiscal subsidies and carbon emissions, meaning these subsidies failed to relieve the financing constraints of these enterprises.

This study had some limitations that need to be explored further in subsequent research. First, the impact of fiscal subsidies on carbon emissions was studied, but these subsidies were not compared with other government measures in order to determine which could better help governments realize maximal green development effects. 

Second, the impact of fiscal subsidies on green development exhibited industry heterogeneity. These subsidies had significant governance effects on the carbon emissions of mining enterprises, but impacts on other industries need to be analyzed and addressed in future research. 

Third, in this work, the relationship between fiscal subsidies and carbon emissions was analyzed generally, while the impact of specific fiscal policies on carbon emissions was analyzed in depth. In fact, these fiscal policies were designed to remedy losses, support corporate innovations, and assist industry development. 

## Figures and Tables

**Table 1 ijerph-19-16256-t001:** Descriptive statistics.

*Variable*	Mean	Median	Max	Min	SD	N
*Sub*	0.0043	0.0017	0.9646	0	0.0194	3230
*C*	10.5550	10.6387	11.9297	5.3519	0.7512	3230
*ROA*	0.0359	0.0313	1.5315	−1.2335	0.0824	3230
*Growth*	0.1661	0.0811	8.9790	−0.8282	0.4207	3230
*Lev*	0.4869	0.4961	2.8487	0.0071	0.2058	3230
*Size*	22.6787	22.5589	28.6365	15.5564	1.5317	3230
*Board*	0.3684	0.3333	0.7143	0.1111	0.0512	3230
*Ggxc*	15.1386	15.1275	18.9415	0	0.8759	3230
*Ls*	0.3914	0.3817	0.8999	0.0339	0.1670	3230
*Pt*	0.1737	0	1	0	0.3788	3230

**Table 2 ijerph-19-16256-t002:** Pearson correlation analysis.

*Variable*	*Sub*	*C*	*ROA*	*Growth*	*Lev*	*Size*	*Board*	*Ggxc*	*Ls*	*Pt*
*Sub*	1									
*C*	−0.014	1								
*ROA*	0.011	0.010	1							
*Growth*	0.021	0.000	0.246 ***	1						
*Lev*	0.021	−0.018	−0.374 ***	−0.154 ***	1					
*Size*	−0.146 ***	−0.058 ***	0.000	−0.081 ***	0.332 ***	1				
*Board*	−0.082 ***	−0.062 ***	−0.038 **	−0.038 **	−0.019	0.080 ***	1			
*Ggxc*	−0.093 ***	−0.076 ***	0.151 ***	−0.051 ***	−0.022	0.443 ***	0.001	1		
*Ls*	−0.025	−0.056 ***	0.074 ***	−0.077 ***	0.135 ***	0.409 ***	0.059 ***	−0.036 **	1	
*Pt*	0.055 ***	0.016	−0.008	0.080 ***	−0.150 ***	−0.243 ***	0.031 *	−0.072 ***	−0.176 ***	1

Note: *, **, and *** denote the significance levels at 10%, 5%, and 1%, respectively.

**Table 3 ijerph-19-16256-t003:** Benchmark regression results.

*Variable*	(1)	(2)
*Sub*	−4.4844 ** (−1.99)	−5.6254 ** (−2.45)
*ROA*		0.5795 ** (2.42)
*Growth*		0.0231 (0.52)
*Lev*		−0.0182 (−0.24)
*Size*		−0.0008 (−0.06)
*Board*		−0.6935 *** (−2.76)
*Ggxc*		−0.1050 *** (−5.11)
*Ls*		−0.2983 *** (−3.31)
*Pt*		0.0054 (0.16)
Control variables	No	Yes
Constant	10.4351 *** (148.18)	12.3641 *** (39.91)
*R^2^*	0.0462	0.0965
*N*	3230	3230

Note: **, and *** denote the significance levels at 10%, 5%, and 1%, respectively.

**Table 4 ijerph-19-16256-t004:** Regression results of different models.

*Variable*	Fixed-Effects Model	Robustness Test	First Diff	
			(1)	(2)
*Sub*	−5.6254 ** (−2.45)	−5.6254 *** (−2.72)	−7.4671 *** (−2.76)	−7.8907 *** (−2.91)
*ROA*	0.5795 ** (2.42)	0.5795 ** (2.39)		0.6168 (1.53)
*Growth*	0.0231 (0.52)	0.0231 (0.66)		−0.0093 (−0.13)
*Lev*	−0.0182 (−0.24)	−0.0182 (−0.31)		−0.3438 *** (−2.71)
*Size*	−0.0008 (−0.06)	−0.0008 (−0.10)		0.0372 * (1.69)
*Board*	−0.6935 *** (−2.76)	−0.6935 *** (−3.75)		0.4559 (1.06)
*Ggxc*	−0.1050 *** (−5.11)	−0.1050 *** (−7.50)		−0.1535 *** (−4.51)
*Ls*	−0.2983 *** (−3.31)	−0.2983 *** (−5.54)		−0.3693 ** (−2.41)
*Pt*	0.0054 (0.16)	0.0054 (0.19)		0.0627 (1.15)
Control variables	Yes	Yes	No	Yes
Constant	12.6848 *** (39.91)	12.3641 *** (90.27)	0.1359(0.99)	1.7644 *** (3.35)
*R^2^*	0.0666	0.0965	0.0687	0.0838
*N*	3230	3230	2326	2326

Note: *, **, and *** denote the significance levels at 10%, 5%, and 1%, respectively.

**Table 5 ijerph-19-16256-t005:** Regression results of different periods.

*Variable*	2009–2013	2009–2020
*Sub*	−6.5230 ** (−2.08)	−2.6568 * (−1.84)
*ROA*	0.5766 * (1.69)	0.2643 (1.40)
*Growth*	0.0132 (0.27)	0.0219 (0.66)
*Lev*	0.0089 (0.07)	−0.1070 (−1.35)
*Size*	0.0229 (1.09)	0.0057 (0.42)
*Board*	−0.7940 * (−1.87)	−0.6315 ** (−2.34)
*Ggxc*	−0.1246 *** (−3.90)	−0.1057 *** (−4.93)
*Ls*	−0.1112 (−0.74)	−0.2926 *** (−2.98)
*Pt*	0.0446 (0.76)	0.0060 (0.16)
Constant	12.0871 *** (24.10)	12.3456 *** (36.60)
*R^2^*	0.1071	0.0871
*N*	1015	2952

Note: *, **, and *** denote the significance levels at 10%, 5% and 1%, respectively.

**Table 6 ijerph-19-16256-t006:** Regression results for the role of green-technology innovation.

*Variable*	*Sub* *→* *C*	*Sub* *→* *Green*	*Sub* *→* *Green* *→* *C*
*Sub*	−5.6080 ** (−2.44)	8.4342 *** (4.25)	−4.4237 * (−1.94)
*Green*			−0.1404 *** (−6.89)
*ROA*	0.6052 ** (2.52)	−0.5135 *** (−2.47)	0.5331 ** (2.23)
*Growth*	0.0204 (0.46)	−0.0604 (−1.58)	0.0120 (0.27)
*Lev*	−0.0156 (−0.21)	−0.3649 *** (−5.57)	−0.0669 (−0.88)
*Size*	−0.0022 (−0.17)	0.2718 *** (24.24)	0.0360 *** (2.57)
*Board*	−0.6823 *** (−2.70)	0.2614 (1.20)	−0.6456 *** (−2.58)
*Ggxc*	−0.1073 *** (−5.20)	−0.0154 (−0.86)	−0.1095 *** (−5.35)
*Ls*	−0.2880 *** (−3.18)	0.3703 *** (4.73)	−0.2360 *** (−2.61)
*Pt*	0.0073 (0.22)	0.0078 (0.26)	0.0084 (0.25)
Constant	12.4141 *** (39.88)	−6.1106 *** (−22.71)	11.5560 *** (34.68)
*R^2^*	0.0959	0.3716	0.1092
*N*	3212	3212	3212

Note: *, **, and *** denote the significance levels at 10%, 5% and 1%, respectively.

**Table 7 ijerph-19-16256-t007:** Regression results of the role of financing constraints.

*Variable*	*Sub* *→* *C*	*Sub* *→* *SA*	*Sub* *→* *SA* *→* *C*
*Sub*	−5.5732 ** (−2.36)	1.1797 (1.59)	−5.0961 ** (−2.17)
*SA*			−0.4044 *** (−7.22)
*ROA*	0.6253 *** (2.58)	−0.2874 *** (−3.78)	0.5090 ** (2.11)
*Growth*	0.0208 (0.47)	0.0841 *** (6.04)	0.0549 (1.24)
*Lev*	−0.0076 (−0.10)	−0.2629 *** (−11.01)	−0.1139 (−1.48)
*Size*	−0.0033 (−0.26)	0.0677 *** (16.60)	0.0240 * (1.79)
*Board*	−0.7158 *** (−2.84)	0.2561 *** (3.24)	−0.6122 ** (−2.45)
*Ggxc*	−0.1074 *** (−5.20)	−0.0340 *** (−5.24)	−0.1212 *** (−5.89)
*Ls*	−0.2902 *** (−3.20)	0.2702 *** (9.48)	−0.1810 ** (−1.98)
*Pt*	0.0080 (0.23)	0.0267 *** (2.51)	0.0188 (0.56)
Constant	12.4393 *** (40.00)	−4.5635 *** (−46.68)	10.5937 *** (26.45)
*R^2^*	0.0972	0.4222	0.1118
*N*	3212	3212	3212

Note: *, **, and *** denote the significance levels at 10%, 5% and 1%, respectively.

## Data Availability

Not applicable.

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
