# Peer review of "The Impacts of Fiscal Subsidies on the Carbon Emissions of Mining Enterprises: Evidence from China"

_ijerph, 2022, doi:10.3390/ijerph192316256_

Round 1

Reviewer 1 Report

The paper presents the impact of fiscal subsidies on the carbon emissions of mining enterprises: evidence from China. It is a topic of interest to the researchers in the related areas and the paper needs minor improvement before acceptance for publication. My detailed comments are as follows:

1.    It is noted that your manuscript needs minor editing by someone with expertise in technical English editing paying particular attention to English grammar, and sentence structure so that the goals and results of the study are clear to the reader.

2.    The conclusion should be concise and only summarize the most important contribution of the research.

Author Response

Point 1: It is noted that your manuscript needs minor editing by someone with expertise in technical English editing paying particular attention to English grammar, and sentence structure so that the goals and results of the study are clear to the reader.

Response 1: Thank you for the suggestion. We have modified this expression throughout the article(including abstract)according to the comment and the amendments are highlighted in red in the revised manuscript.

Point 2: The conclusion should be concise and only summarize the most important contribution of the research.

Response 2: Thank you for the suggestion. In Section 5, the conclusion has been rephrased according to the comment in the revised manuscript(Lines 375-398).

Reviewer 2 Report

This paper investigates the impact of fiscal subsidies on the CO emissions of Chinese mingning enterprises by using panel regression method. The topic is meaningful under the macroscopic background of 'double carbon' target in the context of China. However, I have some questions on this paper.

1. Line 79-81, the authors concluded that the the western and Chinese researchers focus on different perspectives of the policies and measures' influences on carbon emissions. However, the literature review part (Line 47-78) does not have a sufficient review by Chinese research and western countries research.

2. Some of the references of the literature review part are not related to the topic of this paper. For example, the No. 15 paper (Ter-Mikaelian et al.'s paper) pay attention to the forest biological energy's impacts on carbon emissions. Does this paper have any relavance to the mining companies? I do not think so.

3. 'Carbon emission' mentiones in the former studies often indicate CO2 (carbon dioxide) emission. Why you select dependent variable CO (carbon monoxide) in your research?

4. Line 171-184, to better uncover the inpacts of fiscal subsidies on carbon emissions, this paper controls several variables that also have importancee influence on carbon emissions. Please clarify the references you refer to. 

5. The equation 4 (Line 284) introduces green innovation technology to explore its impacts on carbon emissions. You select approved green patents number as the green innovation techonology. As we all know, there is time lag between the patents' approvement and real using. Anyways, the 'Green' and 'CO' in Equation 4 are the same time slice varible. How do you think about the time lag?

Finally, I suggest major revision of the paper.

Author Response

Response to Reviewer 2 Comments

Point 1: Line 79-81, the authors concluded that the the western and Chinese researchers focus on different perspectives of the policies and measures' influences on carbon emissions. However, the literature review part (Line 47-78) does not have a sufficient review by Chinese research and western countries research.

Response 1: Thank you for underlining this deficiency. This section was revised and modified according to the information showed in the work suggested by the reviewer(Lines 29-70).

Point 2: Some of the references of the literature review part are not related to the topic of this paper. For example, the No. 15 paper (Ter-Mikaelian et al.'s paper) pay attention to the forest biological energy's impacts on carbon emissions. Does this paper have any relavance to the mining companies? I do not think so.

Response 2: This phrase was modified according to the comment (Lines 60-63).

Point 3: 'Carbon emission' mentiones in the former studies often indicate CO2 (carbon dioxide) emission. Why you select dependent variable CO (carbon monoxide) in your research?

Response 3: I'm sorry for the misunderstanding. 'Carbon emission' mentioned in our studies is CO2 (carbon dioxide) emission. This sentence was rephrased according to the comment (Line 174).

Point 4: Line 171-184, to better uncover the impacts of fiscal subsidies on carbon emissions, this paper controls several variables that also have importance influence on carbon emissions. Please clarify the references you refer to.

Response 4: This sentence was rephrased according to the comment. We have clarified the references refer to(Lines 180-194).

Point 5: The equation 4 (Line 284) introduces green innovation technology to explore its impacts on carbon emissions. You select approved green patents number as the green innovation techonology. As we all know, there is time lag between the patents' approvement and real using. Anyways, the 'Green' and 'CO' in Equation 4 are the same time slice varible. How do you think about the time lag?

Response 5: Thank you for the suggestion.

Due to the availability of data, only approved green patents number can be selected to measure the green innovation of mining enterprises. Because the approved green patents time lags, it is more appropriate to use the number the patents' real using, but it is difficult to obtain the number the patents' real using from open channels. Considering the impact of approved green patents number with a time lag on the carbon emission of mining enterprises, the regression shows that the green patents with a time lag are still significantly negative, with the regression coefficient of -0.0500, T-value of -2.15 and P-value of 0.031, which is significant at the 5% level. It can be seen that even considering the time lag, green innovation technology is still significantly negatively correlated with the carbon emission of miningenterprises.The results can be seen as follows:

Reviewer 3 Report

This paper finds that fiscal subsidies can decrease the carbon emissions of mining enterprises by enhancing the green technology innovation ability of these enterprises, but not by financing constraints. Although the structure of this article is relatively complete, I think there are the following problems should be solved before its publication:

1. Literature review is supposed to be organized and written in a certain order rather than simply listed without logic. I suggest reorganizing the research review part of this article and trying to change different sentence patterns when quoting the author. In addition, the Chinese literature citation of the whole article is too much, so the proportion of references in Chinese should be reduced and the percentage of foreign and recent literature should be increased. In addition, relative studies can be reviewed, e.g., The effects of nurturing pressure and unemployment on carbon emissions: Cross-country evidence; Analysis of energy-related CO2 emissions in China’s mining industry: Evidence and policy implications; The win-win ability of environmental protection and economic development during China's transition; etc.

2. At the end of the introduction, you should use a few words to briefly summarize the arrangements of this article so that readers can have a general understanding of this article’s layout at the beginning.

3. The second paragraph of the Hypothesis Proposed in this paper is confusing since many irrelevant contents have been written. For example, the geographical and income factors mentioned in this paragraph are not involved in the empirical part of the following text. Please pay attention to the difference between the hypothesis and the literature review. The hypothesis part should focus on explaining the theoretical basis of the hypothesis to support your hypothesis, rather than listing some people's different views.

4. There is an extra full stop in line 135. Please pay attention to the details.

5. The correlation coefficients in Table 2 are all less than 0.5, which is more convincing than less than 0.8.

6. In 4.2 and 4.3, you should specify which variables are fixed in the fixed effect model and the method of robustness test. Besides, in Table 4 and Table 5, some data is the same, so please consider whether to merge the two tables.

7. The sentence in Lines 334-341 is too long to understand. Please divide it into several sentences.

8. Information regarding the funder and the funding number should be provided. Please check the accuracy of funding data and any other information carefully.

9. Please ensure that all individuals included in this section have consented to the acknowledgment.

Author Response

Point 1: Literature review is supposed to be organized and written in a certain order rather than simply listed without logic. I suggest reorganizing the research review part of this article and trying to change different sentence patterns when quoting the author. In addition, the Chinese literature citation of the whole article is too much, so the proportion of references in Chinese should be reduced and the percentage of foreign and recent literature should be increased. In addition, relative studies can be reviewed, e.g., The effects of nurturing pressure and unemployment on carbon emissions: Cross-country evidence; Analysis of energy-related CO2 emissions in China’s mining industry: Evidence and policy implications; The win-win ability of environmental protection and economic development during China's transition; etc.

Response 1: Thank you for underlining this deficiency. This section was revised and modified according to the information showed in the work suggested by the reviewer(Lines 29-71). The relative studies and references have been cited in our study as you refer to.

Point 2: At the end of the introduction, you should use a few words to briefly summarize the arrangements of this article so that readers can have a general understanding of this article’s layout at the beginning.

Response 2: Thank you for the suggestion. We have briefly summarized the arrangements of this article at the end of the introduction(Lines 72-75).

Point 3:The second paragraph of the Hypothesis Proposed in this paper is confusing since many irrelevant contents have been written. For example, the geographical and income factors mentioned in this paragraph are not involved in the empirical part of the following text. Please pay attention to the difference between the hypothesis and the literature review. The hypothesis part should focus on explaining the theoretical basis of the hypothesis to support your hypothesis, rather than listing some people's different views.

Response 3: Thank you for underlining this deficiency. We have Modified throughout the text according to the comment (Lines 77-152).

Point 4: There is an extra full stop in line 135. Please pay attention to the details.

Response 4: This phrase was modified according to the comment.

Point 5: The correlation coefficients in Table 2 are all less than 0.5, which is more convincing than less than 0.8.

Response 5: We have modified the correlation coefficients in Table 2 are all less than 0.5(Line 226).

Point 6: In 4.2 and 4.3, you should specify which variables are fixed in the fixed effect model and the method of robustness test. Besides, in Table 4 and Table 5, some data is the same, so please consider whether to merge the two tables.

Response 6: We have specified which variables are fixed in the fixed effect model and the method of robustness test based on OLS (Lines 231-249). Please see in Table3 and Table4.

Besides, We have merged the Table 4 and Table 5 into Table4 as one table(Line 266).

Point 7: The sentence in Lines 334-341 is too long to understand. Please divide it into several sentences.

Response 7: We have modified this expression throughout the text according to the comment(Lines 331-345).

Point 8: Information regarding the funder and the funding number should be provided. Please check the accuracy of funding data and any other information carefully.

Response 8: We have modified and checked the accuracy of funding data and any other information carefully according to the comment(Line 403).

Point 9: Please ensure that all individuals included in this section have consented to the acknowledgment.

Response 9: This sentence was rephrased according to the comment (Lines 407-408).

Round 2

Reviewer 2 Report

The authors answered most of my questions. However, the introduction section in this version still cannot provide the audience a good map of the main topic. The current version only illustrate the importance of the research on fiscal policies' impacts on carbon emission. Why you choose mining enterprises as your  case? The authors should further include the importance of the mining industry. The literature review should also supplement a paragraph about the fiscal policies' impacts on the carbon emission of mining sector.

Author Response

Response to Reviewer Comments

Point 1: The authors answered most of my questions. However, the introduction section in this version still cannot provide the audience a good map of the main topic. The current version only illustrate the importance of the research on fiscal policies' impacts on carbon emission. Why you choose mining enterprises as your case? The authors should further include the importance of the mining industry. The literature review should also supplement a paragraph about the fiscal policies' impacts on the carbon emission of mining sector.

Response 1: Thank you for underlining this deficiency. Why you choose mining enterprises as your case? The authors should further include the importance of the mining industry.

This section was revised and modified according to the comment (Lines 24-34).

The literature review should also supplement a paragraph about the fiscal policies' impacts on the carbon emission of mining sector.

According to the comment, the literature review should also supplement a paragraph about the fiscal policies' impacts on the carbon emission of mining sector in Section 2 (Lines 83-148).Please check them.

Furthermore, this article has undergone English language editing by MDPI. The text has been checked for correct use of grammar and common technical terms, and edited to a level suitable for reporting research in a scholarly journal.

Reviewer 3 Report

Accept.

Author Response

We would like to express our sincere thanks to the reviewers for the constructive and positive comments.

Thank you very much